# Updates on Biodegradable Formulations for Ocular Drug Delivery

**DOI:** 10.3390/pharmaceutics15030734

**Published:** 2023-02-22

**Authors:** Ta-Hsin Tsung, Yi-Hao Chen, Da-Wen Lu

**Affiliations:** Department of Ophthalmology, Tri-Service General Hospital, National Defense Medical Center, Taipei 11490, Taiwan

**Keywords:** biodegradable drug delivery, ocular drug delivery, biodegradable polymers, nanoparticle drug delivery, polymeric micelles, liposomes, hydrogels, biodegradable implants

## Abstract

The complex nature of the ocular drug delivery barrier presents a significant challenge to the effective administration of drugs, resulting in poor therapeutic outcomes. To address this issue, it is essential to investigate new drugs and alternative delivery routes and vehicles. One promising approach is the use of biodegradable formulations to develop potential ocular drug delivery technologies. These include hydrogels, biodegradable microneedles, implants, and polymeric nanocarriers such as liposomes, nanoparticles, nanosuspensions, nanomicelles, and nanoemulsions. The research in these areas is rapidly growing. In this review, we provide an overview of recent updates in biodegradable formulations for ocular drug delivery over the past decade. Additionally, we examine the clinical use of different biodegradable formulations in various ocular diseases. The aim of this review is to gain a deeper understanding of potential future trends in biodegradable ocular drug delivery systems and to raise awareness of their potential for practical clinical application as a means of providing new treatment options for ocular diseases.

## 1. Introduction

The human eye is a very delicate organ, protected by incredible anatomical and physiological barriers, which makes the eye a highly protected organ from systemic circulation, a so-called immune-privileged organ [1]. The structure of the eye can be divided into two main parts: the anterior segment and the posterior segment. The cornea, conjunctiva, aqueous humor, iris, ciliary body, and lens together make up the anterior segment; while the posterior segment consists mainly of the vitreous humor, sclera, retina, choroid, and optic nerve. Intrinsic and unique barriers in ocular anatomy and physiology protect the eye from environmental toxins and microorganisms, but also limit the access of therapeutic drugs.

Drugs can be delivered to different parts of the eye through various routes, such as topical, systemic, intravitreal, and periocular administration. However, the most patient-friendly routes, such as topical and systemic routes, often have poor bioavailability due to various barriers such as the tear film, cornea, sclera, blood–aqueous barrier, and blood-retinal barrier [2,3,4,5]. Each layer of ocular tissues utilizes specific properties to form different barriers after drug delivery along a certain route of administration. As a result, the development of innovative ocular drug delivery systems (DDSs) that utilize biodegradable formulations for sustained release and increased permeability has emerged as a key area of research.

The eye is a complex organ that is vulnerable to a multitude of vision-threatening conditions, such as glaucoma, dry eye disease, keratitis, conjunctivitis, age-related macular degeneration, diabetic retinopathy, and retinal vascular occlusion. In order to effectively treat these disorders, innovative strategies for ocular drug delivery have been developed, with a particular focus on biodegradable formulations. These biodegradable DDSs mainly include polymeric micelles, polymeric nanoparticles, liposomes, dendrimers, microemulsions, nanosuspensions, implants, microneedles, and hydrogels. These formulations offer a range of benefits, including targeted release of therapeutic agents, increased solubility and stability, enhanced permeability, and extended residence time, all of which contribute to improved drug efficacy. Currently, there are numerous studies examining the therapeutic potential of biodegradable drugs in ocular therapy both in vitro and in vivo. Despite the promise of these drug-loaded biodegradable formulations, their clinical use is limited due to several challenges including difficulties in delivering the drugs to the inner structures of the eye through local infusion, issues with stability, sterilization, low drug loading, high cost, and strong irritation caused by excipients. Therefore, there has been a significant increase in research efforts aimed at addressing these challenges and improving the potential of biodegradable drugs as a treatment option for ocular diseases, leading to a flourishing of studies in this field over the last decade.

In this review, we aim to provide an overview of recent advancements in biodegradable ocular DDSs by summarizing the current evidence for ophthalmological applications reported in the last decade (2013–2022). To the best of our knowledge, the majority of the extant literature on biodegradable DDSs examines a single system exclusively. In this vein, the present work stands out as the first comprehensive review to integrate the latest biodegradable ocular DDSs, whether currently on the market or in research. Each system is analyzed with regard to its design, potential for treating different eye diseases, potential challenges, and prospects for future development.

## 2. Barriers to Effective Ocular Drug Delivery

### 2.1. Barriers in the Anterior Segment

#### 2.1.1. Tear Film

The tear film (Figure 1) can act as a barrier to drug delivery due to the high tear turnover rate and the gelatinous mucus layer. Under physiological circumstances, the tear flow is 1–3 μL/min, renewing the tear film every 5 min [6]. However, after topical administration, the increase in volume leads to reflex stimulation and increased tear secretion, so the drug is diluted and easily washed to the nasolacrimal duct by tear fluid, which ultimately leads to poor bioavailability of topically administered drugs [7]. Furthermore, the tear film consists of an outer lipid layer, a middle aqueous layer, and an inner mucus layer. Mucin in the tear film forms a hydrophilic layer on the glycocalyx of the ocular surface, which protects the eye from cellular debris and pathogens, forming another barrier for drug absorption [8,9].

#### 2.1.2. Cornea

The cornea is one of the major barriers to topical drug absorption (Figure 1). It is about 500 μm thick and is a transparent, avascular tissue composed of five layers including the epithelium, Bowman’s layer, stroma, Descemet’s membrane, and endothelium [2,10]. The outermost apical layer of the corneal epithelium is surrounded by the intercellular tight junctions, which act as a barrier to the diffusion of drug molecules through the paracellular pathway by sealing the intercellular space [11]. Furthermore, the corneal epithelium is physiologically negatively charged, so cationic compounds bind and penetrate the cornea more readily than anionic substances [12]. The stroma, which accounts for 90% of the corneal thickness, is composed of highly organized collagen fibers whose highly hydrated structure constitutes an important barrier to the penetration of lipophilic drug molecules [13]. The corneal endothelium, the innermost hexagonal cell monolayer, contributes to the maintenance of aqueous humor and corneal transparency by virtue of its selective carrier-mediated transport and secretory function [14]. These properties make the cornea a major barrier and challenge for drug delivery to the ocular anterior segment.

#### 2.1.3. Blood–Aqueous Barrier

The blood-aqueous barrier (BAB) (Figure 1) consists of the non-pigmented ciliary epithelium of the ciliary body, the endothelial cells of the iris vessels, and the inner lining endothelium of the Schlemm’s canal [4]. The BAB controls the movement of solutes between the anterior and posterior segments through poorly permeable tight junctions. Tight junctions act as gatekeepers, preventing the free passage of molecules from iris blood vessels and blocking the passage of plasma-derived albumin and hydrophilic drugs from entering the aqueous humor. Therefore, intravenous treatment of anterior segment disease is generally considered impractical [15].

### 2.2. Barriers in the Posterior Segment

#### 2.2.1. Vitreous Humor

The vitreous humor is a clear, gel-like substance that fills the space between the lens and the retina (Figure 1) [16]. The vitreous body has a volume of approximately 4 mL and is a highly hydrated three-dimensional structure composed mainly of water (99%) and the remainder consists of non-collagenous proteins; types I, V, IX, XI collagens; hyaluronic acid (HA); proteoglycans of chondroitin sulfate; and heparan sulfate [17]. The primary function of the vitreous body is to maintain the integrity of the eye and transport nutrients between the retina [18]. The viscous gel of the vitreous humor greatly restricts the diffusion of molecules from the vitreous humor to the retina. The large and charged molecules are difficult to transport to the retina on account of aggregation behavior, and may interact with negatively charged HA and anionic collagens, eventually leading to the precipitation of the molecule in the vitreous humor, making drug absorption more difficult [19].

#### 2.2.2. Sclera and Bruch’s–Choroid Complex

The sclera surrounds the outermost layer of the eyeball (Figure 1), connecting the anterior and posterior parts of the eye [20]. The sclera is mainly composed of an extracellular matrix consisting of collagen fibers, proteoglycans, and glycoproteins to maintain the shape of the eyeball and prevent foreign bodies from entering the posterior tissues. Therefore, it is difficult for drugs with high lipophilicity and large molecular weight to permeate through the pores of the aqueous sclera [21]. In addition, the transscleral permeability is also strongly influenced by the charge of the molecules. In contrast to penetration across the cornea, positively charged molecules are less permeable across the sclera than negatively charged molecules because the proteoglycan matrix of the sclera is negatively charged, causing positively charged solutes to bind and impede their passage through the tissue [22].

The choroid, a pigmented middle layer between the sclera and retina, is a highly vascularized coating that covers most of the posterior outer portion of the eye. The choroid consists of a network of fenestrated capillaries that supply oxygenated and nutritious blood to the outer retina and the retinal pigment epithelium (RPE) layer; it is supported by Bruch’s membrane, a 2 to 4 μm thick elastic membrane that also represents the basement membrane of the RPE [23,24]. The Bruch’s–choroid (BC) complex poses a more critical obstacle to transscleral drug delivery because it is more discriminating than the sclera itself, especially for positively charged lipophilic drugs, since the solute interacts with the tissue, forming sustained-release drug depots in the BC complex [25].

#### 2.2.3. Blood-Retinal Barrier

The blood-retinal barrier (BRB) is a physiological barrier that primarily regulates the movement of proteins, ions, and water from the systemic circulation to the retina (Figure 1). The BRB is composed of internal and external components. The inner BRB consists of tight junctions between retinal capillary endothelial cells, and the outer BRB consists of tight junctions between retinal pigment epithelial cells located between photoreceptors and choroidal capillaries [26]. The tight junctions between these cells allow the BRB to selectively protect the retina from foreign substances in the bloodstream, effectively limiting the transport of molecules, especially hydrophilic compounds and macromolecules, from the choroidal blood circulation to the posterior segment of the eye [3,27]. These ocular static and dynamic barriers limit the penetration of the administered drug into the ocular cavity.

**Figure 1 pharmaceutics-15-00734-f001:**
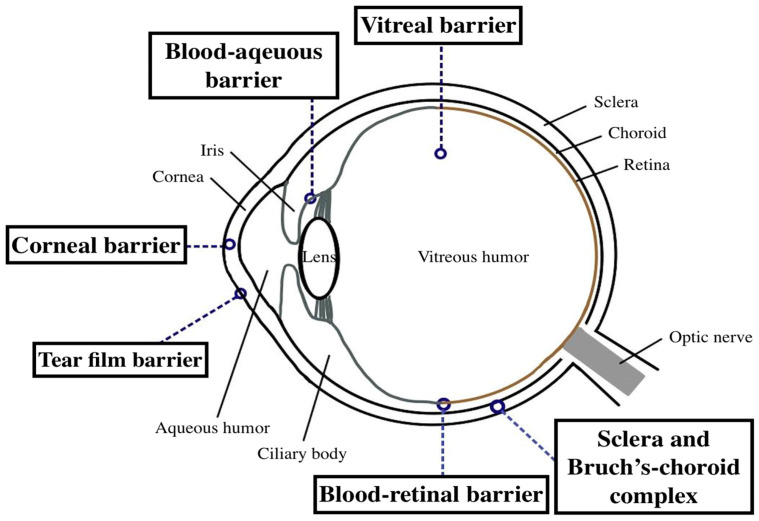
Schematic illustration of the ocular structure with the ocular barriers. The ocular barriers in the anterior segment include the tear film, cornea, and blood-aqueous barrier. The ocular barriers in the posterior segment include the vitreous humor, sclera, Bruch’s–choroid complex, and blood-retinal barrier.

## 3. Conventional Routes of Ocular Drug Delivery

### 3.1. Topical Administration

Drug delivery by topical routes usually involves conventional dosage forms such as solutions, gels, ointments, and suspensions, which account for about 90% of commercially available ophthalmic formulations, and the usual sites of action are the cornea, conjunctiva, sclera, and other tissues of the anterior segment such as the iris and the ciliary body [1,28]. These conventional formulations are usually administered by the topical route to treat common anterior segment diseases such as conjunctivitis, keratitis, dry eye disease and glaucoma to achieve better patient compliance [29]. In most cases, the administered drug is absorbed through the corneal and conjunctival routes. After the previously mentioned barriers of tear dynamics and the corneal barrier, less than 5% of drugs are successfully absorbed through the ocular surface [30]. Therefore, after the installation of topical eye drops, the low bioavailability makes it difficult to produce therapeutic drug concentrations in the deeper posterior segment ocular tissues [31].

### 3.2. Systemic Administration

Considering the higher vasculature compared to the retinal capillaries, drugs readily enter the choroid after intravenous or oral systemic administration [32]. The fenestrated choriocapillaris are highly permeable, allowing drug molecules to achieve a rapid equilibration between the blood circulation and the extravascular space of the choroid [23]. Systemic drug delivery to the retina is theoretically feasible, but in practice it remains a challenge because the BAB and BRB act as the major barriers to anterior and posterior segment drug delivery, respectively. Limited accessibility to the targeted ocular segments restricts the effectiveness of systemic administration, requiring higher dosage to achieve clinically significant therapeutic effects, which may lead to systemic side effects. Therefore, specific oral or intravenous targeting systems are required to deliver molecules through the choroid to deeper layers of the retina, while taking into account both therapeutic efficacy and safety [32,33].

### 3.3. Periocular and Intraocular Injections

Periocular injection delivery methods include subconjunctival, subtenon, peribulbar, retrobulbar, and subretinal administration, while intraocular injections are generally performed via intracameral and intravitreal injection. Because of the complexity of the ocular anatomy and physiology, drug delivery to the posterior segment of the eye is still a great challenge in clinical practice. Therefore, the periocular and intraocular routes of administration are currently used in an attempt to overcome the inefficiency of topical and systemic administration in delivering therapeutic drug concentrations to the posterior segment [22,32]. Intravitreal injection is the most common and widely recommended route of drug administration to treat posterior ocular diseases, and it can be an effective way to deliver high therapeutic concentrations. However, due to the frequent need for highly invasive injections to maintain therapeutic levels, it can lead to low patient compliance and an increased risk of complications such as endophthalmitis, ocular hypertension, cataracts, vitreous hemorrhage, and even retinal detachment [34].

## 4. Novel Biodegradable Ocular Drug Delivery Systems

To overcome ocular drug delivery barriers, improve ocular bioavailability, and maintain constant drug levels at the site of action, several advances have been made in the past few decades. Biodegradable ocular DDSs are methods of delivering drugs to the eye designed to break down and be eliminated from the body through natural metabolic processes [35,36]. These systems can help sustain drug release over an extended period of time and reduce the need for frequent injections or other forms of administration, which can make treatment more patient-friendly, increase patient compliance, and help reduce the risk of complications.

### 4.1. Nanotechnology-Based Systems

Nanotechnology-based drug delivery systems are particle-based systems that can be used to deliver drugs, proteins, genetic material, or other therapeutics to specific cells or tissues in the body [37]. In recent years, researchers have been investigating new ocular nanosystems (ONSs) that utilize nanotechnology to improve the bioavailability of ocular drugs.

#### 4.1.1. Nanoparticles

Nanoparticles (NPs) are colloidal particles with a spherical shape that are typically characterized by a size range spanning 1 to 1000 nm [38]. NPs are classified into two groups: polymeric NPs and lipid NPs. Biodegradable polymeric NPs are solid colloidal particles made from either natural resources such as chitosan, alginate, albumin, and dextran or synthetic macromolecules including polylactic acid (PLA), polyglycolide (PGA), poly lactic-co-glycolic acid (PLGA), polyaspartic acid, poly-alkyl cyanoacrylate, and polyethyleneimine (PEI) [36,38]. Polyethylene glycol (PEG) and polyvinylpyrrolidone (PVP) are hydrophilic polymers that serve as mucoadhesive nanosystems to increase the residence time of NPs in ocular mucosal and anterior corneal tissues [39]. Polymeric NPs are hampered by cytotoxicity and a lack of suitable large-scale production techniques, whereas lipid NPs offer a lower risk of toxicity by relying on biodegradable and non-toxic lipid components. Solid lipid nanoparticles (SLNs) and nanostructured lipid carriers (NLCs) are composed of a solid matrix that allows for controlled drug release, while also being more stable and cost-effective. SLNs utilize physiological lipids, eschew organic solvents, and enable large-scale production, thereby leading to increased drug bioavailability, better drug protection, and improved control of drug release. NLCs are formed by regulating the mixing of solid lipids and liquid oil, creating unique nanostructures in the matrix. The limitations of SLNs, such as limited drug-loading capacity and drug expulsion during storage, can be addressed through the use of NLCs [40].

According to current studies [41,42,43,44,45], drug-loaded NPs have the advantages of high drug retention, low dosing frequency, and low toxicity for the treatment of ocular surface diseases, and have the potential to replace conventional formulations as the main choice of anterior ocular disease treatment in the near future. However, although polymeric and lipid NPs have a strong carrying capacity for fat-soluble substances, their carrying capacity for water-soluble substances is insufficient, which is a direction that needs to be studied and improved.

#### 4.1.2. Liposomes and Niosomes

Liposomes are spherical, double-membraned structures composed of phospholipids and cholesterol [46]. Phospholipids, such as phosphatidylcholine, phosphatidylserine, and phosphatidylethanolamine, possess lipidic and surfactant-like properties that aid in binding to the ocular surface and permeating through pores [47]. Cholesterol is incorporated to maintain lipid bilayer rigidity and to prevent drug leakage. The amphiphilic nature allows for lipophilic drugs to be incorporated between their double-layered membranes, while hydrophilic drugs can be encased in the central aqueous compartment [48]. Niosomes, a subtype of non-ionic surfactant-based vesicles, are composed of a lipid bilayer similar to liposomes, but exhibit greater stability during formation and storage due to the incorporation of non-ionic surfactants and cholesterol as excipients [49].

Liposomes have advantages in drug delivery, such as a range of environments for drug cargoes to attach [50], high drug loading capacity, and the ability to modify the surface with different molecules to tailor pharmacokinetic profiles [51,52]. Liposomes interact with cells through various means, and their cell membrane-like structure makes them biocompatible and degradable, leading to reduced drug toxicity and sustained drug release [53,54]. However, there have been several challenges in using liposomes, such as low drug loading, leakage of embedded drugs, poor long-term stability, high cost of production, difficulty with sterilization, and low formulation targeting. These challenges highlight the need for the development of new liposome preparations that can overcome these limitations and improve the treatment of ocular diseases.

#### 4.1.3. Dendrimers

Dendrimers are novel polymeric nanoarchitectures that are characterized by their three-dimensional, highly branched, and highly symmetric structure. They were first synthesized using a divergent synthesis approach by Fritz Vögtle in 1978, and their unique properties, including nanoscopic size, narrow polydispersity index, tunable surface functionality, availability of multiple functional groups on the surface that hydrophilic and lipophilic drugs can be conjugated to, and high surface area-to-volume ratio, make them an attractive choice for ocular drug delivery [55,56]. Polyamidoamine (PAMAM) dendrimers are the most extensively studied, and are commercially available due to their well-defined structure and precise size control [57]. However, the use of dendrimers in clinical applications is limited by concerns about cytotoxicity, particularly for cationic dendrimers, which have a higher affinity for cell membranes and are more cytotoxic and hemolytic than anionic and neutral dendrimers. The cytotoxicity is also dependent on the generation of the dendrimer, with higher generations generally exhibiting greater toxicity [58]. To address cytotoxicity concerns, researchers are investigating methods for reducing the toxicity, such as modifying their surface chemistry or using biocompatible polymers. However, changes to the surface properties may also affect the overall properties and effectiveness of the dendrimer for the intended application [59]. Despite these challenges, the unique properties of dendrimers continue to make them a promising option for ocular drug delivery.

#### 4.1.4. Microemulsions

The concept of microemulsions (MEs) wasn’t widely acknowledged until Hoar and Schulman’s 1943 publication. MEs have small droplet sizes (typically between 5 and 200 nm) and are characterized by low surface tension and high spreading coefficient. MEs are colloidal nanodispersions consisting of oil, aqueous phase, surfactant, and co-surfactant [60,61], which can be either oil-in-water or water-in-oil types depending on their relative proportions. The surfactants lower the surface tension between the oil and water phases, allowing for dispersion and creating a protective barrier that maintains stability [62]. These properties make MEs a thermodynamically stable system that can enhance drug absorption and penetration, particularly for poorly water-soluble drugs, and improve drug delivery to the posterior segment of the eye [63].

MEs have potential applications in ocular drug delivery due to their small droplet size, phase transition behavior, and ease of application compared to traditional eye drops [64]. However, their practical application is limited by several factors. One of the main concerns is the droplet size, as smaller droplets can be more easily retained in the eye and provide a longer duration of drug action. However, smaller droplets also increase the likelihood of irritation and toxicity. Additionally, long-term stability can be an issue, and the large number of surfactants and co-surfactants used in their preparation can also cause irritation. Therefore, when developing ophthalmic formulations using MEs, it is important to consider their potential for irritation.

#### 4.1.5. Nanosuspensions

Nanosuspensions (NSs) are a type of biphasic system composed of colloidal dispersions of nanoscale poorly water-soluble drug particles, with a diameter of less than 1 µm, suspended in an aqueous medium, and are suitable for ocular administration of poorly water-soluble drugs [65]. NSs exhibit a distinct contrast to conventional matrix-framework nanosystems in that they do not necessitate carrier materials. They consist solely of drug NPs in the nanoscale size range, and can be stabilized through the inclusion of certain excipients, such as surfactants, viscosity enhancers, and charge regulators [66]. When a drug is made into an NS, the conversion of the drug reduces its particle size, increasing specific surface area and saturation solubility. This improves bioavailability in topical administration by increasing contact area, residence time, and therapeutic concentration in the tissue, which leads to more effective treatment and potentially lower doses [67,68].

NSs are effective for drug delivery, but surfactants used as suspending agents can cause irritation and toxicity. To address this, researchers are exploring alternative methods, such as encapsulating NSs in gel matrices, bioadhesive matrices, or ophthalmic implants for sustained drug release and reduced risk. However, NSs are less stable than other colloidal particle systems, requiring careful consideration and optimization of toxicity and stability for ocular drug delivery systems.

#### 4.1.6. Polymeric Micelles

In recent years, using polymeric micelles (PMs) as a drug delivery method has gained significant attention in the treatment of ocular diseases. These micelles are composed of amphiphilic polymers, which contain both hydrophobic and hydrophilic segments and form spherical structures when added to a solvent such as water. The hydrophobic segments aggregate at the core of the micelle, encapsulating hydrophobic drugs, while the hydrophilic segments extend outwards and interact with the solvent, increasing drug solubility [69,70]. The unique properties of PMs offer a promising alternative to traditional drug delivery systems, as they can efficiently transport drugs to their targeted site.

The selection of the core-forming polymer is a crucial aspect in PM design, as it influences stability, drug loading capacity, and drug release characteristics [71]. Two commonly used polymers for PM synthesis are PLGA and chitosan (CH). PLGA is a biodegradable and biocompatible polymer that degrades in a controlled manner, and can be combined with polyvinyl alcohol (PVA) to improve drug release. CH is a natural cationic polysaccharide derived from chitin, with excellent mucoadhesion and penetration properties ideal for drug delivery in mucosa and ophthalmic areas [72]. PMs offer several benefits, including improved penetration through lipophilic corneal epithelial and endothelial cells due to their amphiphilic nature and small size, enhanced contact with the ocular surface due to their mucoadhesive properties, and the ability to create clear aqueous solutions for easy application in eye drops without causing vision interference [73]. However, PMs have limitations, such as low drug loading, difficulty in controlling release rates, and challenges in large-scale production. Additionally, their potential toxicity to eye tissue requires further study.

The novel ONSs discussed above have been demonstrated to possess superior pharmacokinetic and pharmacodynamic properties compared to conventional therapies, as evidenced by their controlled release of therapeutic drugs, targeted delivery to specific tissues, extended contact time, prolonged circulation, enhanced penetration through biological barriers, and decreased elimination rates [69,74,75]. One of the major challenges is the optimization of drug loading and release, which could be achieved through the incorporation of various types of functional materials or the modification of the nanoparticle surface. Another important challenge is the improvement of the stability and biocompatibility of nanosystems, especially for long-term use. Additionally, the manufacturing process and scale-up production of nanosystems remain a challenge due to the complex synthesis procedures and the lack of cost-effective production techniques. Despite their potential benefits for ocular drug delivery, several challenges must be addressed to improve their efficacy and clinical feasibility. One of the major challenges is achieving adequate drug loading and release, which may be hindered by drug stability and solubility, as well as the properties of the nanosystem itself. Another obstacle is optimizing the size and surface properties of the nanosystem to ensure effective penetration and targeting in the eye while minimizing toxicity and immune reactions. Furthermore, regulatory and manufacturing challenges must be overcome to enable large-scale production and clinical translation of these complex drug delivery systems. Future research efforts in the field of ONSs could focus on several aspects to further enhance their efficacy and address current limitations. One such aspect is the development of more efficient and eco-friendly synthesis methods, which can improve the reproducibility and scalability of nanosystems. Advanced techniques such as 3D printing and microfluidics can also be explored to achieve this objective. Additionally, the design of nanocarriers, including size, shape, and surface properties, can be optimized to enhance their penetration across ocular barriers and improve their drug delivery efficiency. Co-delivery of multiple drugs or therapeutic agents using nanosystems can also be explored, as it has the potential to enhance therapeutic efficacy and reduce the frequency of administration. These research areas could lead to the development of more effective and targeted ocular drug delivery systems with reduced side effects and improved patient compliance.

### 4.2. Biodegradable Microneedles

Minimally invasive microneedle-based ocular drug delivery is a cutting-edge technology that can revolutionize drug administration to the eye. Initially, solid and hollow microneedles (MNs) were developed for transdermal drug delivery [76]. Recently, this technology has been adapted for the ocular surface to avoid complications associated with conventional needles [77]. Micron-sized needles (25–2000 μm in height) reduce patient discomfort and allow for precise localization of drug formulations [78,79]. The utilization of non-degradable materials such as stainless steel, ceramics, and aluminum oxide in the production of MNs has been motivated by their ease of fabrication and sterilization [80]. However, non-degradable MNs pose a significant limitation due to their non-biocompatible nature, leading to chronic inflammation and foreign body reactions that can cause long-term tissue damage, hence reducing their clinical application. Biodegradable MNs made from polylactic acid, polyglycolic acid, PLGA, methacrylate hyaluronic acid (MeHA), and CH have been considered as promising alternatives for ocular and other applications. Drugs are enclosed within the MNs, and the biodegradable polymer matrix that degrades over time releases the encapsulated drug in a controlled way. Numerous drug delivery systems, including gel formulations and nanoparticle suspensions, have been evaluated for use with microneedles [81,82].

The clinical implementation of MNs for ocular drug delivery remains a challenge due to the limited long-term efficacy caused by the rapid dissolution of biodegradable composites. This limitation is particularly problematic for the treatment of chronic ocular diseases, which require sustained drug delivery over a prolonged period. It is also important to note that the distribution of drugs to the posterior segment of the eye after intrascleral or suprachoroidal microneedle injection may be limited by previously mentioned barriers. Consequently, further research is needed to investigate the injection and retraction forces, intraocular pressure, and overall safety of the technology, as well as the long-term delivery of therapeutic proteins to comprehensively understand the potential benefits of MNs for ocular applications [77].

### 4.3. Hydrogels

Hydrogels, which are water-swollen network structures, have potential applications in drug delivery and tissue engineering due to their high water content and mechanical properties that mimic the extracellular matrix and soft tissues [83]. Hydrogel properties can be tuned to mimic biological matrices, and exhibit similar properties such as high hydrophilicity, high hydration, and closely matched mechanical properties [84]. These characteristics may also preserve the physicochemical state of biological drugs over extended periods, especially when compared to more rigid, hydrophobic polymer matrices that are commonly used for drug encapsulation and release, and can result in the degradation of peptides and proteins. Hydrogels have recently gained prominence as ophthalmic formulations due to their properties, and in situ gels and ocular implants are two common subtypes.

#### 4.3.1. In Situ Gels

In situ gels, also known as “smart hydrogels,” transition from a liquid to a gel state in response to physiological signals, such as changes in temperature or pH, after application in vivo [85]. This creates a bioadhesive network that extends drug retention and facilitates sustained release, reducing the need for frequent dosing and improving patient adherence. This ultimately improves patient adherence to the treatment regimen. In situ gel systems can be administered via various routes and have been demonstrated to be effective in serving as a vehicle for the delivery of drug-laden NPs, NSs, NEs, and liposomes for the treatment of ocular disorders [86,87]. Solid in situ gelling formulations, such as polymeric films [88] and electrospun nanofibers [89], have also been studied as ophthalmic DDSs. Solid gel formulations have potential advantages such as ease of handling and self-administration, improved storage stability, and enhanced drug bioavailability.

#### 4.3.2. Hydrogel Implants

Hydrogels have a long history of utilization in ocular implants, such as punctal plugs and contact lenses, and can serve as versatile drug delivery vehicles for ocular surface disorders [90]. Punctal plugs are frequently utilized in dry eye disease management, but drug depletion and low bioavailability have been observed [91]. Encapsulating the drug in nanocapsules or nanomicelles prior to embedding them in punctal plugs has been explored to improve drug delivery efficacy [92,93]. Contact lenses provide an alternative for ocular surface diseases treatment by immersing the lenses in a drug solution, but rapid drug release and visual acuity issues are associated with this method [94,95,96]. Ring-implanted contact lenses that encapsulate drugs within NPs and deliver them through a ring-shaped boundary region have been developed to mitigate these issues. This approach has demonstrated prolonged ocular retention and sustained release of hyaluronic acid [97]. Nanoparticle-encapsulating contact lenses have the potential to enable the development of innovative ocular therapies.

### 4.4. Biodegradable Implants

Currently, the Food and Drug Administration (FDA) has only approved intraocular implants as a delivery vehicle for sustained release of small molecular drugs administered intravitreally to the retina. Implants can be biodegradable or non-biodegradable, with the latter demonstrating superior precision in regulating drug release and prolonged release durations. However, they require surgical intervention for implantation and removal. Biodegradable polymer-based implants with PLGA, PLA, or PGA gradually release the drug in a controlled manner and do not require surgical removal [98]. Several commercial implant options available for the treatment of ocular diseases approved by the FDA vary in active ingredient and release profile, such as Trivaris^®^ (triamcinolone acetonide suspension; Allergan, Inc., Irvine, CA, USA), Kenalog^®^ (triamcinolone acetonide suspension; Bristol-Myers Squibb, Princeton, NJ, USA), Iluvien^®^ (fluocinolone acetonide non-biodegradable implant; Alimera Sciences, Inc., Alpharetta, GA, USA), Ozurdex^®^ (dexamethasone Implant; Allergan, Inc., Irvine, CA, USA), Durysta^®^ (bimatoprost implant; Allergan, Inc., Irvine, CA, USA), Dexycu^®^ (dexamethasone intraocular suspension; Icon Bioscience Inc., Newark, CA, USA), and others. Among these options, only Ozurdex^®^, Dexycu^®^ and Durysta^®^ are biodegradable implants. Ozurdex^®^ was approved in 2009 for the treatment of macular edema associated with retinal vein occlusion and has a release profile of up to six months [99]. Dexycu^®^ is the first and only FDA-approved, single-dose, sustained-release, intracameral steroid on the market as of 2018. It provides anti-inflammatory efficacy beginning from day 1 to day 30 with a single injection after cataract surgery. In 2020, Durysta^®^ received approval for use as an intracameral injection to reduce intraocular pressure (IOP) in patients with open angle glaucoma or ocular hypertension for a period of 4 to 6 months. Additionally, there are novel photosensitive biodegradable implants under development, such as OcuLief™ and EyeLief™ by Re-Vana Therapeutics Ltd. (Belfast, UK) As the field of ocular disease treatment evolves, it is likely that there will be an increased emphasis on the development of biodegradable implant technology.

## 5. Application of Different Biodegradable Ocular Drug Delivery Systems on Ocular Diseases

The anterior and posterior segments of the eye are vulnerable to vision-threatening ailments, including glaucoma, uveitis, and ocular surface diseases for the former, and age-related macular degeneration, diabetic retinopathy, and retinal vascular occlusion for the latter. Biodegradable drug delivery systems are being explored as a potential therapy for ocular disorders, as they provide targeted and sustained release of therapeutic agents, increasing drug effectiveness.

### 5.1. Dry Eye Disease

According to the latest revised second version of the International Dry Eye Workshop report of the Tear Film and Ocular Surface Society, released in 2017 [100], dry eye disease (DED) is defined as a multifactorial disorder of the ocular surface, characterized by destabilization of the tear film and accompanied by ocular symptoms. Tear film instability and hyperosmolarity, ocular surface inflammation and injury, and neurosensory abnormalities play a crucial role in the etiology of DED [101]. The management of DED necessitates the utilization of multiple medication forms, including lubricants such as artificial tears and sodium hyaluronate eye drops, anti-inflammatory agents such as corticosteroid and nonsteroidal anti-inflammatory drugs (NSAIDs), immunosuppressants such as cyclosporine A (CsA) and tacrolimus, and other pharmaceuticals such as secretory agents and autologous serum eye drops [101,102].

Corticosteroids are widely used anti-inflammatory drugs known for their efficacy in managing eye inflammation. To enhance bioavailability and minimize systemic side effects, the development of biodegradable corticosteroid formulations for anterior segment diseases is ongoing (Table 1). Several investigations have reported successful administration of corticosteroids, namely prednisolone acetate, dexamethasone sodium phosphate, fluorometholone, and triamcinolone acetonide, through distinct nanosystems [42,43,44,103,104,105,106]. Tan et al. have developed a chitosan thermosensitive hydrogel with dexamethasone-loaded NLCs which can be administered as an eye drop and transform into hydrogel upon contact with the conjunctival sac [105]. The release study indicated the sustained release of dexamethasone, but is limited to in vitro experiments and lacks in vivo results. Soiberman et al. developed a subconjunctival injectable dendrimer-dexamethasone gel that was found to be more effective than free-Dex in attenuating corneal inflammation, as evidenced by reduced macrophage infiltration and pro-inflammatory cytokine expression [104]. The gel exhibited prolonged efficacy for 2 weeks following a single injection, and improved corneal thickness and clarity without increasing IOP. However, the study’s design was limited to 14 days and a single injection, raising concerns about potential clinical application. Hanafy et al. developed self-assembled prednisolone acetate-loaded NP hydrogels using chitosan and sodium alginate as counter-ions [42]. In female guinea pig eyes, the gel demonstrated superior anti-inflammatory effects compared to a micronized drug-loaded gel. However, comparing these formulations may not be ideal as they have different properties and mechanisms of action. A comparison to a conventional topical ophthalmic aqueous solution or suspension would provide a more relevant and practical comparison. Considering that, Ryu et al. put forward a novel approach involving the incorporation of dexamethasone-loaded PLGA NPs into a quickly dissolving dry tablet for ocular administration [44]. In contrast to the established commercial dexamethasone eye drop Maxidex^®^, the aforementioned strategy resulted in sustained drug release for a duration of 10 h and a 2.6-fold boost in ocular drug bioavailability. Notwithstanding, the limitation of this study pertains to its commercial application, since the shelf life of the dry film is constrained, and the manufacturing process is relatively intricate.

NSAIDs, such as nepafenac, ibuprofen, indomethacin, pranoprofen, and flurbiprofen, encapsulated in NPs, NSs, microemulsions, and liposomes, resulted in an improved precorneal residency time and increased ocular bioavailability as listed in Table 1. For example, indomethacin-loaded microemulsions and nanoemulsions have been proposed as an alternative delivery system to the topical application due to their ability to enhance corneal penetration and reduce ocular irritation in a rat model [107]. However, the long-term effectiveness and safety of the formulations are not evaluated, since the assessment of the efficacy was limited to a duration of 5 h in the study. Additionally, encapsulation of another NSAID, dexibuprofen, in PLGA nanoparticles has been demonstrated to enhance its release and efficacy in the treatment of ocular inflammation in a rabbit model [41].

Topical CsA is an effective treatment for DED for its anti-inflammatory properties, with several FDA-approved non-biodegradable formulations available, such as Restasis^®^ (CsA 0.05%, Allergan Inc, Irvine, CA, USA), Ikervis^®^ (CsA 1 mg/mL, Santen Pharmaceutical, Osaka, Japan), and Cequa^®^ (CsA 0.09%, Sun Pharma, Cranbury, NJ, USA) [91,108,109]. In the past decade, biodegradable drug delivery systems have been studied to enhance the therapeutic effect of CsA (Table 1). Luschmann et al. found that CsA-loaded NSs and micelles were able to maintain higher concentrations in the corneal tissue of rabbits compared to Restasis^®^ [110]. Nevertheless, the study’s duration was limited to 180 min, which renders the long-term safety and efficacy of the formulations uncertain. Similarly, Yan et al. demonstrated the efficacy of CsA-loaded cationic NSs in therapeutic concentrations to anterior ocular tissues via topical drop instillation [111]. An ophthalmic CsA in-situ gel based on LNCs developed by Eldesouky et al. was shown to prolong ocular retention and enhance tissue penetration [112]. The study identified a potential limitation in its inability to demonstrate a significant difference in the effectiveness of the 0.1% and 0.05% LNC formulations using the Schirmer tear test, despite both showing similar results to the marketed CsA-NE under the same conditions. The rabbit model used in this study may not accurately represent severe cases of dry eye that would require the higher concentration of 0.1%.

**Table 1 pharmaceutics-15-00734-t001:** Studies of biodegradable formulations for anterior segment diseases.

Study	Drug and System	Experimental Models	Description
Chen et al., 2022 [113]	Tacrolimus-loaded cationic liposomes	Male New Zealand rabbits	The use of cationic liposomes to encapsulate FK506 prolonged ocular retention, increased corneal FK506 levels, and reduced reactive oxygen species and dry eye–related inflammation factors.
Han et al., 2022 [114]	Polyhedral oligomeric silsesquioxane hybrid thermoresponsive FK506 hydrogel	Female C57BL/6 mice	This hydrogel possesses good biocompatibility, prolonged ocular retention and enhanced therapeutic efficiency in comparison with Commercial FK506 in dry eye.
Mirgorodskaya et al., 2022 [107]	Indomethacin-loaded MEs and nanoemulsions	A carrageenan-induced edema rat model	The Indomethacin nanoemulsion showed a prolonged release and slowed down the progression of carrageenan-induced edema compared to the unencapsulated drug.
Akbari et al., 2021 [97]	Hyaluronic acid-loaded chitosan nanoparticle-containing ring-implanted PVA contact lens	In vitro	The ring-implanted contact lens showed sustained release of hyaluronic acid for up to 14 days, and a cellular study indicated no corneal epithelial cell toxicity.
Eldesouky et al., 2021 [112]	CsA lipid nanocapsules as thermoresponsive in situ gel	Male New Zealand colored rabbits	This drug delivery form extends the ocular stay of CsA and improves its tissue penetration capacity. A subsequent pharmacodynamic study showed it was more effective than the commercially available cyclosporine nanoemulsion in increasing tear production in rabbits.
Ma et al., 2021 [115]	Levocarnitine thermosensitive in situ gel	New Zealand rabbits	The formulation can significantly increase the amount of tear secretion and the number of conjunctival goblet cells, improve the degree of corneal damage and the pathological morphology of the lacrimal gland, and down-regulate the apoptosis rate of corneal epithelial cells.
Yan et al., 2021 [111]	CsA loaded cationic NSs	Male adult New Zealand albino rabbits	The cationic NSs can deliver CsA to anterior ocular tissues in effective therapeutic concentrations (10–20 μg/g) with topical drop instillation.
Nagai et al., 2020 [116]	Rebamipide solid nanoparticle-based sustained-release formulations	Adult rabbits	The rebamipide formulation showed sustained release compared to the commercial suspension, and improved mucin levels and tear film breakup in an N-acetylcysteine treated rabbit model.
Sánchez-López et al., 2020 [41]	Dexibuprofen-loaded PLGA NPs	New Zealand albino rabbits	These formulations were able to release dexibuprofen more effectively for the treatment of ocular inflammation.
Hanafy et al., 2019 [42]	Prednisolone acetate loaded chitosan-deoxycholate self-assembled NPs	Female guinea pig eyes	The formulation achieved a twofold increase in the prednisolone release after 24 h when compared with the commercial micronized drug-loaded gel.
Wang et al., 2019 [43]	Dexamethasone sodium phosphate-loaded NPs using zinc ion bridging with dense coatings of polyethylene glycol	A corneal neovascularization rat model	A single subconjunctival administration inhibited corneal angiogenesis in rats within 2 weeks, without increasing IOP or causing toxicity at the injection site, which could be an effective strategy for preventing and treating corneal neovascularization.
Ryu et al., 2019 [44]	Dry tablets dexamethasone-loaded micelles	Male New Zealand white rabbits	This formulation increased 2.6-fold the ocular drug bioavailability when compared to Maxidex^®^.
Gonzalez-Pizarro et al., 2019 [117]	Fluorometholone loaded PLGA NPs in situ forming gels	New Zealand albino male rabbits	The formulation administration improved precorneal residency time, leading to increased ocular bioavailability and penetration into deep tissues such as aqueous humor and crystalline.
Tatke et al., 2018 [106]	Ion-sensitive in situ gelling system containing triamcinolone acetonide–loaded solid lipid NPs	New Zealand albino rabbits	The ex vivo on rabbit corneas showed improved permeability with the formulation compared to simple solutions. The gelation increased drug absorption and prolonged drug residence time on the ocular surface and in the conjunctiva sac, leading to sustained release and minimal pre-corneal drug loss.
Ren et al., 2018 [118]	Azithromycin–cholesteryl hemisuccinate ion pair in liposome	A dry eye rat model	Azithromycin liposomes showed enhanced corneal permeation compared to the azithromycin solution.
Huang et al., 2018 [119]	Gelatin–epigallocatechin gallate NPs with hyaluronic acid	New Zealand white rabbits	The eye drops effectively prolonging drug retention on the ocular surface and effectively inhibiting ocular inflammation in dry eye rabbits.
Vaidehi et al., 2017 [120]	Tacrolimus loaded modified liposomes with propylene glycol	New Zealand albino rabbits	Topical application of the formulation in rabbits showed prolonged precorneal retention, improved corneal and non-corneal penetration, and increased intraocular drug levels compared to a drug solution.
García-Millán et al., 2017 [103]	Triamcinolone acetonide-loaded nanosuspensions with Poloxamer 407 and PVA as stabilizing agents	Polyhydroxyethyl methacrylate soft contact lenses	The NSs significantly improved drug loading and release in soft contact lenses compared to a drug-saturated solution.
Soiberman et al., 2017 [104]	Subconjunctival injectable dendrimer-dexamethasone gel	Rats and rabbits	A subconjunctival injection of dexamethasone gel prolonged efficacy for 2 weeks and showed improved outcomes with reduced central corneal thickness, improved corneal clarity, and no elevation in IOP.
Tan et al., 2017 [121]	Nanostructured lipid carriers-based chitosan thermosensitive hydrogel with dexamethasone	In vitro	This formulation can be administered in the eye in solution state by dropping, and will transform to hydrogel when it contacts with the conjunctival sac due to its thermosensitivity. The results of the release study showed a sustained release of dexamethasone in vitro.
Zeng et al., 2016 [122]	Tacrolimus loaded hyaluronic acid-coated niosomes	New Zealand albino rabbits	Hyaluronic acid-coating improved the adhesion to mucin, and the formulation resulted in increased precorneal retention, pharmacokinetics, and ocular bioavailability of tacrolimus.
Chen et al., 2016 [123]	Flurbiprofen-loaded chitosan liposomes	New Zealand albino rabbits	The formulation could prolong pre-corneal retention and improve transcorneal penetration compared to flurbiprofen-solution without ocular damage or abnormal clinical signs.
Cholkar et al., 2015 [124]	CsA-loaded nanomicelles	New Zealand White albino rabbits	The instillation of nanomicelles resulted in the highest concentration of CsA in the anterior chamber, but a higher level was detected in the retina, indicating their potential to deliver drugs to the posterior segment through a conjunctival-scleral pathway.
Addo et al., 2015 [125]	Albumin–chitosan microparticles with 0.66% atropine sulfate	Rabbits	The formulation had longer contact time and superior effects on mydriasis in rabbits than the standard 1% atropine sulfate solution.
Abrego et al., 2014 [126]	Pranoprofen-loaded PLGA nanoparticles with PVA as a stabilizer	In Vitro	The formulation showed sustained release of the drug compared to commercial eye drops and free drug, with optimal ocular tolerance as no irritation reactions were detected within 5 min of the assay.
Yu et al., 2014 [127]	Injectable in situ polyethylene glycol hydrogels for bevacizumab	In Vitro	The developed hydrogel showed no cytotoxicity in vitro after 7 days and sustained the release of encapsulated bevacizumab for 14 days, which might have potential to treat the corneal neovascularization.
Kesavan et al., 2013 [128]	Mucoadhesive chitosan-coated cationic microemulsion of dexamethasone	New Zealand White rabbits	The formulation was developed to treat chronic uveitis and showed stability for 3 months in vitro, with improved therapeutic effect of the incorporated steroid in vivo.
Luschmann et al., 2013 [110]	CsA-loaded nanosuspensions and micelle	the rabbit cornea	The formulation had significantly higher drug concentrations in the corneal tissues of rabbits compared to the commercially available Restasis^®^ group.

PVA, polyvinyl alcohol; CsA, cyclosporin A; NS, nanosuspension; PLGA, Poly Lactic-co-Glycolic Acid; NP, nanoparticle; IOP, intraocular pressure.

Tacrolimus is another widely used immunosuppressant for DED treatment. Biodegradable formulations of tacrolimus, such as cationic liposome [113] and thermoresponsive in situ gel formulations [114] have been developed to improve ocular retention, enhance corneal tacrolimus concentration, and decrease levels of reactive oxygen species and inflammatory factors associated with DED (Table 1).

### 5.2. Conjunctivitis and Keratitis

Conjunctivitis, characterized by inflammation of the conjunctiva, is a prevalent ocular surface disease that can be caused by infectious agents or non-infectious factors, such as allergens, toxins, immune-mediated processes, and neoplastic processes [129]. Keratitis results from the inflammation of the cornea and can be classified as either infectious or non-infectious based on the etiological agent, with bacterial, protozoal (Acanthamoeba), fungal, and viral keratitis being the subcategories of the infectious form [130].

Ocular pharmacological interventions for conjunctivitis and keratitis involve targeting specific causes with antibiotics, antivirals, antifungals, and anti-inflammatory drugs. However, their limited water solubility and short residence time on the ocular surface pose a challenge. To improve therapeutic efficacy, biodegradable formulations have been studied to extend drug release (Table 1 and Table 2). The utilization of a liposomal delivery system loaded with ciprofloxacin was investigated by Taha et al. [131] and Al-Joufi et al. [132] in a rabbit model. The studies revealed that this ciprofloxacin-loaded liposome formulation exhibited enhanced performance as compared to the available commercial product with respect to elimination rate constant, corneal permeability, and relative bioavailability. Nonetheless, the studies did not encompass the evaluation of the safety and toxicity of the formulation, which is a critical aspect in the development of a new drug delivery system, solely focusing on pharmacokinetic parameters.

Polymeric NPs are commonly utilized for delivering hydrophobic drugs in ocular infections. Ameeduzzafar et al. formulated chitosan-based NPs for delivering levofloxacin, which demonstrated biocompatibility for topical ophthalmic use and showed a longer retention time in the ocular area compared to levofloxacin solution [133]. However, the study did not compare the developed formulation with other commercially available formulations. Moreover, it only assessed the antibacterial activity against two bacterial strains (*P. aeruginosa* and *S. aureus*), and the efficacy against other strains was not evaluated.

Roy et al. developed a microneedle ocular patch (MOP) containing amphotericin B (AmB) to treat fungal keratitis [134]. The MOPs showed a significant reduction in Candida albicans load in ex vivo and in vivo infection models. However, the article has limitations as it did not compare the effectiveness of MOPs with other ocular drug delivery systems. Additionally, the long-term stability and shelf life of the MOPs were not addressed. Despite these limitations, this study provides a new direction for the treatment of ocular infections, and further clinical studies are necessary to evaluate the safety and efficacy of MOPs in humans.

**Table 2 pharmaceutics-15-00734-t002:** Studies of biodegradable formulations for ocular infection.

Study	Drug and System	Experimental Models	Description
Al-Joufi et al., 2022 [132]	Ciprofloxacin loaded liposome	Male New Zealand white albino rabbits	This formulation exhibited superior performance in comparison to the commercially available product, Ciloxan^®^, in terms of peak aqueous humor concentration, time to reach peak aqueous humor concentration, elimination rate constant, corneal permeability, and relative bioavailability.
Abbas et al., 2022 [135]	Oxytetracycline-loaded gelatin-polyacrylic acid NPs laden in situ gelling solution	White albino rabbits	The optimized formulation was tested for its ability to combat Pseudomonas aeruginosa, a common cause of keratitis, both in vitro and in vivo on a rabbit eye conjunctivitis model. The results showed sustained effectiveness against keratitis and comparable antibacterial activity to a commercial product.
Mahboobian et al., 2020 [136]	thermosensitive in situ gel nanoemulsions containing acyclovir	Male New Zealand albino rabbits	The sustained release pattern of the drug was observed in the formulation compared to the control solution. The drug permeation of the optimal formulation was about 2.8 times higher than the control solution.
Gugleva et al., 2019 [137]	Doxycycline hyclate niosomes	In vitro	In vitro release studies revealed a sustained release profile. Additionally, the encapsulation efficiency and particle size of the niosomes were found to be physically stable after being stored for 2 months at 4 °C.
Roy et al., 2019 [134]	Amphotericin B containing polymeric microneedle ocular patch with PVA and PVP	New Zealand White male rabbits infected with Candida albicans	The microneedles resulted in a significant reduction of the Candida albicans load within the cornea, as determined through both ex vivo and in vivo infection models.
Ameeduzzafar et al., 2018 [133]	Levofloxacin loaded chitosan NPs	New Zealand albino rabbits	The formulated levofloxacin possessed superior antibacterial activity against *P. aeruginosa* and *S. aureus*, as well as reduced corneal clearance and nasolacrimal drainage, resulting in increased retention of the drug in comparison to a simple solution.
Xie et al., 2017 [93]	A hyaluronic acid-based in situ punctal plug containing ofloxacin-loaded microcapsules	In vitro	The development of a one-step in-situ drug-encapsulation process was revealed in the study, which enables the creation of a resorbable hydrogel punctal plug with extended drug release.
Silva et al., 2017 [138]	Chitosan/sodium TPP-hyaluronic acid-based NPs containing ceftazidime	In vitro cell lines	The produced NPs interact with mucin and increase the residence time of the NPs on the eye surface, which improves the drug absorption and reduces the frequency of administration with no toxicity.
Kalam et al., 2016 [139]	Gatifloxacin-loaded microemulsion	New Zealand white rabbits	The optimized microemulsions were stable and exhibited improved adherence to the cornea, leading to an increased diffusion of gatifloxacin into the anterior chamber. This resulted in a twofold increase in gatifloxacin concentration compared to a conventional dosage form.
Kapanigowda et al., 2015 [140]	Ganciclovir chitosan microspheres	Male and female Wistar rats	The formulation demonstrated a significant increase in maximum concentration when compared to a ganciclovir solution. The in vivo ocular pharmacokinetic studies in conjunction with the histopathology report showcased the effectiveness and tolerability of the formulation.
Sharma et al., 2015 [141]	Amikacin sulphate laden polymeric NPs	Male New Zealand albino rabbits	The ocular bioavailability of the formulation was greater than that of currently available eye drops, and did not cause any discomfort to the cornea for up to 12 h after administration.
Silva et al., 2015 [142]	Daptomycin-loaded chitosan NPs	In vitro	The in vitro release of daptomycin was found to be complete within 4 h. The bacteria remained susceptible to daptomycin encapsulated in NPs. The addition of mucin was found to enhance their mucoadhesive properties for endophthalmitis.
Taha et al., 2014 [131]	Ciprofloxacin-loaded liposomes	Male New Zealand white Albino rabbits	The formulation revealed superior aqueous humor concentrations and a threefold increase in bioavailability compared to the commercially available eye drops (Ciprocin^®^)
Üstündag-Okur et al., 2014 [143]	Ofloxacin-loaded microemulsion	Male New Zealand rabbits	The use of this microemulsion as a treatment for bacterial keratitis was found to be noninferior to a commercial formulation containing 0.3% Ofloxacin.
Mudgil et al., 2013 [144]	Moxifloxacin-loaded PLGA nanosuspension	Freshly excised goat eyes	The formulation demonstrated improved transcorneal permeation and prolonged microbial efficacy against *S. aureus* and *P. aeruginosa*, compared with the marketed eye drop Moxicip^®^.

NP, nanoparticle; PVA, polyvinyl alcohol; PVP, polyvinyl pyrrolidone; PLGA, Poly lactic-co-glycolic acid.

### 5.3. Uveitis

The uveal tract, which consists of the iris, the ciliary body, and the choroid, can be subject to inflammation. Based on the primary site of inflammation, uveitis can be further categorized into anterior, intermediate, posterior, and panuveitis [145]. The treatment of uveitis may encompass anti-inflammatory and corticosteroid agents, either as monotherapy or in conjunction with other immunosuppressants, which to some extent coincide with the management of DED and ocular infections, as shown in Table 1.

In contrast to drugs targeted toward the anterior segment of the eye, those aimed at the posterior uveitis have multiple ocular barriers that must be overcome to reach the intended site of action. Therefore, the use of biodegradable carriers is attracting attention (Table 3). Polymeric nanomicelles have been shown to have the potential for effective drug delivery to the targeted site. The concept of dexamethasone-encapsulated polymeric nanomicelles was introduced by Vaishya et al. [146]. The outcomes of ex vivo permeability studies demonstrated that the rigid nanomicelle core could effectively transport dexamethasone to the posterior segment when administered topically, thus offering a promising treatment approach for intermediate to posterior segment uveitis. Nevertheless, it is crucial to acknowledge that these experiments were conducted in vitro and ex vivo. After that, Xu et al. created the chitosan oligosaccharide-valylvaline-stearic acid nanomicelles incorporating dexamethasone and evaluated its efficacy in rat and rabbit models [147]. The nanomicelles showed a prolonged release pattern, a high level of adhesion to mucosal surfaces, and improved penetration capabilities. However, the study did not provide a comparative analysis of the designed nanomicelles with other existing ocular DDSs, thereby limiting the determination of the superiority of the developed nanomicelles over other formulations. Future research directions should focus on comparative analysis to further explore the potential advantages of the developed nanomicelles.

Wu et al. demonstrated an additional application of nanomicelles through intravitreal injection of rapamycin-loaded polymeric micelles [148]. In rats, these micelles have been shown to retain rapamycin in retinal pigment epithelial cells for at least 14 days, leading to improved therapeutic outcomes for the treatment of autoimmune uveitis compared to the administration of rapamycin suspension alone. However, this study has a limited duration of only 14 days, leaving the potential long-term therapeutic effect or toxicity uncertain. Moreover, the use of a single concentration of rapamycin micelles in this study warrants further investigation on the dose–response relationship in future studies.

**Table 3 pharmaceutics-15-00734-t003:** Studies of biodegradable formulations for posterior segment diseases.

Study	Drug and System	Experimental Models	Description
Tavakoli et al., 2022 [149]	Sunitinib-loaded liposomes	A laser induced CNV mouse model	Intravitreal administration of sunitinib-loaded liposomes showed an inhibitory effect on established neovascularization in a mouse model of laser-induced CNV.
Rudeen et al., 2022 [150]	A hydrogel DDS containing dexamethasone-loaded NPs and aflibercept-loaded microparticles	In vitro	The Combo-DDS hydrogel, consisting of both aflibercept-loaded microparticles and dexamethasone-loaded NPs, showed a sustained release time of 224 days. The swelling ratio and equilibrium water content of Combo-DDS slightly decreased compared to aflibercept-DDS and dexamethasone-DDS.
Wu et al., 2021 [151]	Ovalbumin-encapsulated PLGA NP loaded bilayer dissolving microneedle	Ex vivo with excised porcine sclera	This method of delivering encapsulated proteins has the potential to provide sustained release for over 2 months and effectively bypass the scleral barrier, making it a promising therapy for treating neovascular ocular diseases.
Mehra et al., 2021 [152]	Everolimus loaded nanomicelles prepared using a grafted polymer (Soluplus^®^)	Ex vivo with goat cornea	The formulation is a promising nanocarrier for topical ocular drug delivery for uveitis due to their longer duration in the circulatory system and accumulation in the inflammatory area, as well as their ability to enhance the permeation of the drug through the cornea via the topical route.
Xu et al., 2020 [147]	Chitosan oligosaccharide-valylvaline-stearic acid nanomicelles with dexamethasone	Male rats and male New Zealand albino rabbits	The nanomicelles showed long-lasting release, were well-tolerated, adhered well to mucosal surfaces, and improved penetration.
Blazaki et al., 2020 [153]	Intravitreal injection of calcein, FITC-dextran-4000 and flurbiprofen encapsulated liposome aggregate platform system	Adult pigmented rabbits	The LAP system significantly increased the retention of flurbiprofen in the ocular tissues and decreased inflammatory reactions towards calcein, compared to non-aggregated liposomes.
Chauhan et al., 2019 [154]	Dasatinib encapsulated spray dried PLGA particles	In vitro	The formulation showed sustained release and significant inhibition of collagen matrix contraction in an in vitro scar contraction assay, demonstrating its potential for treating proliferative vitreoretinopathy.
Qiu et al., 2019 [155]	Fenofibrate-loaded PLGA NPs	Male Brown Norway mice	The Feno-NP improved retinal dysfunctions, inhibited retinal leukostasis, diminished retinal vascular leakage, and regulated the over expression of VEGF at eight weeks after the application, with the therapeutic potential for the treatment of DR and nAMD with prolonged drug release and potentially reduced injection frequency.
Lui et al., 2019 [156]	Dexamethasone-loaded PLGA and polyethylenimine NPs with bevacizumab adsorbed onto the surfaces	Male New Zealand White rabbits and male Chinchilla rabbits	These NPs demonstrated a good anti-angiogenic effect and a strong inhibitory effect on VEGF secretion, and is a potential treatment for AMD.
Alami-Milani et al., 2018 [157]	Dexamethasone-loaded polycaprolactone-polyethylene glycol-polycaprolactone micelles	Ex vivo with freshly prepared bovine cornea	The micelles demonstrated improved transcorneal permeation compared to the commercial eye drop, resulting in higher dexamethasone levels in the intraocular tissues after topical administration.
Badiee et al., 2018 [158]	Bevacizumab-loaded chitosan nanoparticles embedded in a hyaluronic acid ocular implant	Rabbit vitreous humor	The results showed that the formulation sustained drug release for 2 months. Using bevacizumab-loaded chitosan NPs within a matrix of hyaluronic acid and zinc cation could be a promising approach for sustained bevacizumab delivery.
Mahaling et al., 2018 [159]	Triamcinolone acetonide-loaded NP with a hydrophobic polycaprolactone core and a hydrophilic Pluronic^®^ F68 shell	A diabetic retinopathy rat model	The NPs decreased retinal inflammation as shown by a reduction in NF-κB, ICAM-1, and TNFα expression after 20 days of treatment. They also reduced glial cell hyperplasia with lower GFAP expression and microvascular complications evidenced by a decrease in VEGF secretion and microvascular tuft formation after 40 days of treatment.
Wu et al., 2016 [148]	Rapamycin-loaded polymeric micelles	A rat experimental autoimmune uveitis model	Retinal pigment epithelial cells in rats retained rapamycin-loaded micelles for at least 14 days after intravitreal injection, extending drug retention time in the retina. The micelle system improved therapeutic outcomes for autoimmune uveitis in rats compared to rapamycin suspension alone.
Adamson et al., 2016 [160]	anti-VEGF molecule loaded microparticles of PolyActive™ hydrogel co-polymer	Primate and rabbit models of wet AMD	The dual domain antibodies (dAb) showed high potency, with a lower IC50 than aflibercept in VEGF receptor binding assays, and retained its activity after being released from microparticles for up to 12 months in vitro. In vivo, the microparticles released functional dual dAb in the eyes of rabbits and primates for up to 6 months, providing sufficient protection against laser-induced grade IV CNV in Cynomolgus.
Yavuz et al., 2016 [161]	Dexamethasone- Polyamidoamine conjugated dendrimers	Male Sprague Dawley rats	Drug-loaded dendrimers enhanced the ocular permeability of dexamethasone after subconjunctival injection, as compared with the free drug.
Varshochianand et al., 2015 [162]	Bevacizumab-loaded albumin PLGA NPs	New Zealand albino rabbits	The prepared NPs provided a sustained-release formulation of bevacizumab with a vitreous concentration of more than 500 g/L, and were extended for about 8 weeks.
Vaishya et al., 2014 [146]	Dexamethasone-encapsulated polymeric nanomicelles	Ex vivo with excised rabbit sclera	Results from ex vivo permeability and rigid nanomicelle core showed that these nanomicelles may be able to deliver dexamethasone to the posterior segment through topical administration, potentially making it a viable option for treating intermediate to posterior segment uveitis.
Luo et al., 2013 [163]	PLGA NPs delivering recombinant Flt23k intraceptor plasmid	Rodent and primate models of CNV	The formulation offers an innovative method of ocular drug delivery through systemic administration that effectively curbs neovascularization and fibrosis in macular degeneration models while overcoming the significant disadvantages of intraocular injection of anti-VEGF agents.
Yandrapu et al., 2013 [164]	Bevacizumab loaded PLGA NPs	A rat model	The in vitro examination of the formulation revealed a sustained release of bevacizumab over a period of 4 months. Upon in vivo evaluation in a rat model, the detection of bevacizumab delivery was observed for a duration of 2 months in the vitreous humor.
Iwase et al., 2013 [165]	Doxorubicin conjugated polyethylene glycol and poly(sebacic acid) NPs	C57BL/6 mice and Dutch belted rabbits	The intraocular injection of NPs (10 μg of doxorubicin) was effective in suppressing neovascularization in transgenic mice that express VEGF in their photoreceptors, resulting in suppression for at least 35 days. The injection of NPs (2.7 mg of doxorubicinin) in rabbits resulted in sustained release with detectable levels in both aqueous humor and vitreous for up to 105 days.

CNV, choroidal neovascularization; DDS, drug delivery system; PLGA, Poly Lactic-co-Glycolic Acid; NP, nanoparticle; LAP, liposome aggregate platform; VEGF, vascular endothelial growth factor; DR, diabetic retinopathy; nAMD, neovascular age-related macular degeneration; NF-κB, nuclear factor-κB; ICAM-1, intercellular adhesion molecule-1; TNF-α, tumor necrosis factor-α; GFAP, glial fibrillary acidic protein.

### 5.4. Age-Related Macular Degeneration (AMD)

AMD is a major contributor to blindness in developed countries, and its complications such as choroidal neovascularization (CNV) and geographic atrophy can be serious and potentially devastating [166]. Intravitreal injection is the preferred method of administering anti-vascular endothelial growth factor (VEGF) drugs or corticosteroids for AMD; however, therapeutic small molecules have poor permeability. The utilization of biodegradable DDSs enhances the permeation of therapeutic agents across biomembranes, thereby improving the treatment of ocular CNV (Table 3).

Liposomes have been studied as a potential drug delivery system for ocular neovascularization. Blazaki et al. developed a novel Liposome Aggregate Platform (LAP) system encapsulated with calcein, FITC-dextran-4000 (FD4), and flurbiprofen. The LAP system increased the retention of flurbiprofen in the posterior segment after intravitreal injection [153]. However, the potential inflammatory response and side effects of the LAP system, which are crucial for its future applications, were not assessed in the study. Tavakoli et al. demonstrated the inhibitory effect of intravitreal administration of sunitinib-loaded liposomes on established neovascularization in a mouse model of laser-induced CNV [149]. A potential drawback of this experiment is its restricted investigation on a mouse model of laser-induced CNV, which may not accurately represent the pathophysiology of CNV in human patients. As a result, future research should focus on examining the effectiveness of liposomal sunitinib in comparison to existing anti-VEGF treatments and its relevance in a clinical setting.

Current research aims to enhance the penetration and prolong the drug action for ocular CNV treatment by combining polymeric NPs with other vehicles. Badiee et al. encapsulated bevacizumab in a chitosan NP, which was incorporated into a hyaluronic acid-based ocular implant [158]. While in vivo experimentation was not conducted, the in vitro studies indicated a prolonged drug release lasting for a duration of two months. Wu et al. presented an example of a ovalbumin-encapsulated PLGA NP-loaded bilayer dissolving microneedle as a method of protein delivery [151]. This approach has the potential to deliver a sustained release of the encapsulated protein for a duration of over 2 months ex vivo and effectively circumvent the scleral barrier. The co-administration of dexamethasone with anti-VEGF agents such as aflibercept and bevacizumab as polymeric NPs has been demonstrated to exhibit a prolonged release profile, a potent anti-angiogenic effect [150,156]. The studies discussed provide a potential avenue for a promising therapeutic approach in treating CNV through the combination of different biodegradable formulations and co-administration of two drugs in a single injection.

### 5.5. Glaucoma

Glaucoma is a major contributor to blindness worldwide and it affects approximately 1% of the global population, with a global age-standardized prevalence of 3.5% among individuals aged 40 years or older [167,168]. The elevated IOP experienced by patients with glaucoma disrupts the dynamic balance of aqueous circulation and results in optic nerve atrophy and visual field defects. The use of topical administration to treat glaucoma is problematic due to several factors, such as poor patient compliance, potential long-term damage to the corneal surface, and low drug bioavailability [169]. Sustained drug delivery could provide a potential solution to these challenges. Antiglaucoma medications, including latanoprost, dorzolamide, brinzolamide, timolol maleate, brimonidine, and pilocarpine, have been explored as formulations in different biodegradable DDSs in various studies, including polymeric nanoparticles [170], liposomes [105,171,172], microneedles [173], and in situ hydrogel systems [174,175,176,177,178,179], demonstrating promising results in terms of bioavailability and sustained release (Table 4).

In March 2020, a remarkable advancement was achieved in the field of glaucoma therapy with the approval of a biodegradable sustained-release IOP lowering implant by the FDA. The implant, known commercially as Durysta^®^ and produced by Allergan plc (Dublin, Ireland), consists of a rod-shaped polymer matrix with 10 µg of bimatoprost, which is gradually released into the eye over a period of several months. The implant aims to address the challenge of non-adherence among glaucoma patients by providing a convenient, long-lasting, and consistent treatment option [180]. The safety and efficacy of the implant were demonstrated by two Phase III clinical trials (ARTEMIS 1 and 2) [181,182]. Results indicated that the majority of patients experienced substantial biodegradation of the implant within 12 months. Furthermore, the implant was effective in lowering IOP, with approximately 80% of patients not requiring additional medication for up to one year after their third implant. Notably, the bimatoprost implant’s clinical use has raised concerns about corneal side effects, with a higher incidence of corneal endothelial cell loss and iritis observed in the implant groups compared to the timolol group. Therefore, the speed of implant degradation and the potential accumulation of debris in the iridocorneal angle warrant further investigation as future targets to improve the safety and efficacy of this innovative DDS.

ENV515 travoprost Extended Release (XR) is a rod-shaped, biodegradable intracameral implant under investigation, designed to provide a steady supply of travoprost for 6 to 12 months. Several studies have confirmed its efficacy. A Phase IIa study compared Travoprost XR implant to topical Travatan Z in 21 glaucoma patients, with the implant group showing a 6.7 mmHg decrease in diurnal IOP by day 25. Another 12-month study was conducted on open-angle glaucoma patients previously treated with prostaglandins. The study compared the IOP reduction in the eyes treated with the implant to that of the fellow eye treated with topical timolol once daily. The results showed a mean IOP reduction of 6.3 mmHg, or 25%, in the study eyes, and was deemed non-inferior to timolol [183,184].

The OTX-TIC intracameral implant is another biodegradable implant that is being researched. It is a biodegradable device from Ocular Therapeutix, featuring a soft hydrogel platform embedded with travoprost-loaded microparticles and a meshwork structure to hold the microparticles. A Phase 1 study assessed the safety, efficacy, durability, and tolerability of the OTX-TIC implant. The study was prospectively designed, multi-center, and open label, with a dose escalation approach. The first 9 subjects in two cohorts showed a reduction in mean IOP from baseline that lasted throughout the 18-month study with no serious adverse events reported [185].

**Table 4 pharmaceutics-15-00734-t004:** Studies of biodegradable formulations for glaucoma.

Study	Drug and System	Experimental Models	Description
Pan et al., 2020 [186]	Dexamethasone and melatonin co-loaded PLGA NPs	A rabbit eye model	The NPs showed sustained release of both drugs in vitro without any burst release. The in vitro cytotoxicity study found no toxicity on R28 cells, similar to the control group. The NPs also showed improved retinal penetration and a significant reduction of IOP.
Roy et al., 2020 [173]	Pilocarpine-loaded microneedle ocular patch using dissolvable PVA and PVPM	Ex vivo with excised human cornea and porcine eye	The patch significantly increased the permeation of pilocarpine across the excised cornea. The availability in the aqueous humor of the porcine eye globe was greater within 30 min of the patch application than the solution formulation.
Agibayeva et al., 2020 [174]	Gellan gum and its 6, 14 and 49% methacrylated derivatives as in situ gelling mucoadhesive formulations of pilocarpine	Chinchilla rabbits	The formulations of pilocarpine hydrochloride that contain gellan gum and methacrylated derivatives improved the drug’s effectiveness. However, the best results were observed with the polysaccharide that had a 6% methacrylation level.
Bhalerao et al., 2020 [175]	Brinzolamide dimethyl sulfoxide in situ gelling solution	New Zealand white rabbits	The tested formulations were found to be safe and effective in reducing IOP, resulting in a decrease from 25–28 mmHg to 12–14 mmHg compared to control samples. Additionally, the test formulations also showed an improvement in the area under change in intraocular pressure from baseline and an extended mean residence time (7.4 to 17.7 h) compared to the commercial suspension of Azopt^®^ (4.9 h).
Arranz-Romera et al., 2019 [187]	Multi-loaded PLGA-microspheres incorporating three recognized neuroprotective agents (dexamethasone, melatonin and coenzyme Q10)	A rodent model of chronic ocular hypertension	In vitro studies showed that multi-loaded microspheres were neuroprotective in a model of glutamate-induced cytotoxicity in R28 cells. In vivo studies found that this formulation provided significant neuroprotection for retinal ganglion cells compared to controls. No neuroprotective effect was observed with empty microspheres or individual single-drug-loaded microspheres.
Orasugh et al., 2019 [176]	Pilocarpine loaded thermo-responsive in situ gelling systems with cellulose nanocrystals	In vitro	The results showed that the formulation had a prolonged release of the drug and lower toxicity.
Franca et al., 2019 [188]	Chitosan/hydroxyethyl cellulose inserts for sustained release of dorzolamide	Male Wistar rats	A single administration of the ocular insert resulted in a significant decrease in IOP for two weeks, while no significant change in IOP was observed in the placebo and untreated groups. The insert also demonstrated a preventative effect on the retinal ganglion cell death.
Sánchez-López et al., 2018 [189]	Memantine loaded PLGA NPs	Morrison’s ocular hypertension model in Dark Agouti rats	In vitro and ex vivo studies showed that NPs provide sustained release and enhanced delivery compared to other formulations. These NPs were also well-tolerated in human retinoblastoma cells and in vivo Draize test. In the rodent model, topical application of the formulation for 3 weeks resulted in a significant reduction of RGC loss.
Fahmy et al., 2018 [171]	Latanoprost/Thymoquinone encapsulated liposome	White albino rabbits	The liposome samples were found to significantly reduce IOP for up to 84 h. Treatment of glaucomatous rabbits with the formulations also improved histopathological lesions in ocular tissue.
El-Feky et al., 2018 [177]	Timolol maleate loaded chitosan-gelatin hydrogel	Male albino rabbits	The hydrogel’s mucoadhesive properties were studied, with in vitro release profiles showing that crosslinking with oxidized sucrose slowed down the release rate of timolol. In vitro and in vivo studies confirmed that the hydrogel sustained timolol release and efficacy for a longer period compared to regular eye drops.
Kouchak et al., 2018 [172]	Dorzolamide loaded-nanoliposome	A randomized control trial in primary open angle glaucoma and ocular hypertension patients	The study measured the effectiveness of dorzolamide-loaded nanoliposome eye drops in reducing IOP, compared to a control group (marketed dorzolamide HCl eye drop). Results showed a significant decrease in IOP in the intervention group, with no significant adverse effects.
Sun et al., 2018 [178]	Gellan gum based brinzolamide ion sensitive in situ gelling system	New Zealand rabbits	The formulation was found to be safe and bioadhesive. The gel formed a strong gel upon contact with simulated tear solutions, enabling the controlled release of brinzolamide.
Morsi et al., 2017 [190]	Nanoemulsion-based ion-sensitive in situ gels containing acetazolamide	In vitro	The formulation demonstrated a prolonged drug release when compared to the plain nanoemulsion. These gels exhibited greater therapeutic effectiveness and a longer-lasting reduction in IOP compared to commercial eye drops and oral tablets.
Salama et al., 2017 [170]	Brinzolamide-loaded PLGA nanoparticles	Male New Zealand Albino rabbits	Injected subconjunctivally in normotensive Albino rabbits, this formulation was able to reduce the IOP for up to 10 days.
Lai et al., 2017 [179]	Intracameral pilocarpine administration with Chitosan-g-poly(N-isopropylacrylamide) in situ gelling delivery system	A rabbit model of experimental glaucoma	The formulation allowed the drug concentration to reach the minimum therapeutic level for treating glaucoma for 42 days during the study. Good ocular biocompatibility with lens epithelial cell cultures was also noted. The effectiveness of pilocarpine in reducing IOP causing miosis and preserving the corneal endothelium was found to be closely related to the drug release profiles.
Tan et al., 2017 [105]	Timolol maleate chitosan coated liposomes	New Zealand white rabbits	The formulation showed a better mucoadhesive effect with a prolonged retention time of the cornea, and an excellent IOP-lowering effect compared with commercial timolol maleate drops.
Sun et al., 2017 [191]	A layered double hydroxide nanoparticle/thermogel composite drug delivery system for sustained release of brimonidine	New Zealand rabbits	The system demonstrated biocompatibility and a lack of cytotoxicity to human corneal epithelial cells. In vivo testing showed sustained drug release from a special contact lens made of this system for at least 7 days, resulting in more effective modulation of IOP relief.
Lavik et al., 2016 [192]	A biodegradable microsphere formulation for timolol maleate	Male New Zealand white rabbits	The use of timolol microspheres in a subconjunctival administration method resulted in a sustained delivery of the drug and a reduction in IOP for up to 90 days in rabbits, without causing any inflammation or toxicity in either the local or systemic areas.
Bravo-Osuna et al., 2016 [193]	Acetazolamide loaded water-soluble mucoadhesive carbosilane dendrimers	New Zealand white rabbits	The eyedrop formulation induced a rapid (within 1 h) and extended (p to 7 h) decrease in IOP. The addition of a small amount of cationic carbosilane dendrimers to an acetazolamide solution was found to be well-tolerated and resulted in an improvement in the drug’s hypotensive effect.
Huang et al., 2016 [194]	Thermosensitive in situ hydrogel of betaxolol hydrochloride	A rabbit model	The in vitro study of the formulation showed an increase in viscosity and a prolonged release of betaxolol hydrochloride. The results of the in vivo study confirmed the improved bioavailability and a significant reduction in IOP.
Lai et al., 2015 [195]	Intracameral pilocarpine administration with gelatin-g-poly(N-isopropylacrylamide) in situ gelling delivery system	A rabbit model of experimental glaucoma	The 2-week in vitro study showed that the formulation was able to provide sustained release of pilocarpine, sufficient for therapeutic action in reducing ocular hypertension. Clinical observations in rabbits also confirmed the effectiveness of the injections through reduction of IOP and preservation of corneal endothelial cell health.
Yu et al., 2015 [196]	Liposome incorporated ion sensitive in situ gels for timolol maleate	New Zealand rabbits	The eye drops were found to be most effective 30 min after administration, with the effect lasting for 240 min. Compared to traditional eye drops, the in situ gels were able to more quickly and effectively lower IOP and had a longer lasting effect.
Mishra et al., 2014 [197]	Acetazolamide loaded poly(propylene imine) dendrimer nanoarchitectures	Normotensive adult male New Zealand albino rabbits	The study revealed that the dendrimer-based formulation prolonged the reduction in IOP to 4 h, compared to the 2-h reduction seen with the acetazolamide solution alone.
Wong et al., 2014 [198]	Liposomal latanoprost	An open-label, pilot study on humans with ocular hypertension or primary open-angle glaucoma	The use of liposomal latanoprost via subconjunctival injection was found to be well tolerated by all six subjects and resulted in a significant decrease in IOP of 47.43% within 1 h and lasting up to 3 months, with a statistically significant reduction observed.
Singh et al., 2014 [199]	Acetazolamide-loaded, pH-triggered polymeric nanoparticulate in situ gel	A rabbit model	Ex vivo study showed higher acetazolamide permeation from this formulation than eye drops and suspension. Nonirritant properties were confirmed by a modified Draize test, and no corneal toxicity was observed. The in situ gel also caused a significant decrease in IOP in rabbits compared to eye drops.
Li et al., 2014 [200]	A brinzolamide drug-resin thermosensitive in situ gelling system	A rabbit model	This stable, non-irritant formulation showed controlled release of brinzolamide over 8 h in vitro. In vivo evaluation revealed improved retention of the drug compared to commercial preparations.
Jung et al., 2013 [201]	Timolol encapsulated nanoparticle loaded silicone-hydrogel contact lenses	Beagle dogs	Incorporating nanoparticles into silicone hydrogels decreases ion and oxygen permeability, increases modulus, and these effects are proportional to the number of nanoparticles used. A gel with 5% nanoparticles can deliver therapeutic doses of timolol for a month with minimal impact on lens properties, as shown in preliminary animal studies in Beagle dogs.

PLGA, Poly Lactic-co-Glycolic Acid; NP, nanoparticle; IOP, intraocular pressure; PVA, polyvinyl alcohol; PVPM, polyvinyl pyrrolidone matrix; RGC, retinal ganglion cell.

Latanoprost FA SR is a biodegradable rod-shaped intracameral implant under development by PolyActiva in Parkville VIC, Australia. It is designed to deliver latanoprost for the treatment of primary open angle glaucoma and is currently in Phase II studies. The primary and secondary efficacy endpoints aim to achieve a 20% reduction in IOP in the low dose cohort. The next advancement in DDSs for the treatment of glaucoma would be the integration of fixed-dose combination drugs in a sustained delivery device. Additionally, a novel sustained delivery system that is coupled with an IOP-monitoring device would also be desirable.

## 6. Conclusions

Over the last decade, significant advances in biodegradable ocular DDSs have been made, with the goal of improving important features such as drug stability, solubility, corneal permeability, and retention time for enhanced performance, bioavailability, patient satisfaction, and compliance. These biodegradable drug delivery technologies are outstanding in their versatility and have vast growth potential. Biodegradable polymers in the creation of these systems offer the potential to revolutionize ocular drug delivery, especially with the development of novel vehicles engineered for controlled and sustained drug delivery to treat vision-threatening diseases. Further studies are essential to understand the mechanisms of action of these systems, ensure quality control and safety, and enhance the efficacy and safety of biodegradable ocular formulations. The development of innovative ophthalmic DDSs remains a future focus and is expected to play a crucial role in improving visual health.

## Data Availability

Not applicable.

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
