# Peer review of "Updates on Biodegradable Formulations for Ocular Drug Delivery"

_pharmaceutics, 2023, doi:10.3390/pharmaceutics15030734_

Round 1
Reviewer 1 Report
I reviewed the manuscript "Updates on biodegradable formulations for ocular drug delivery in recent ten years" by Ta-Hsin Tsung, Yi-Hao Chen, and Da-Wen Lu.
The amount of work done by authors is impressive, the text is well written and the English is fine (I would double check the punctuation but it's just a minor issue).
My first concern is about the lenght of the article. There's nothing wrong with a review of 46 pages, but here a major part of the text is dedicated to simple concepts (e.g. lines 233-234) that are out of the scope of the article. Both section 4 and 5 could be shortened by avoiding the description of concepts that might fit for a book chapter, but not for a review article.
In fact, the crucial information of this article is reported in the tables, where DDSs are presented and summarized, making the paragraphs look dispersive and section 4 almost useless. Moreover, studies reported in the tables are not commented on which limitations authors found or problems linked to the use of those systems, which is a pivotal element when writing a review article. I think authors should replace most of the "basic" information in the paragraphs with a critical comment on the most important studies that they cite.
I think authors should also consider to change perspective, and divide DDSs not by application, but by type (NPs, hydrogels, implants ...) as they did in section 4. The title and the focus of the article are novel formulations for ocular delivery, so this should be the main theme of the whole work.
Author Response
Response to Reviewer 1 Comments
Point 1: My first concern is about the lenght of the article. There's nothing wrong with a review of 46 pages, but here a major part of the text is dedicated to simple concepts (e.g. lines 233-234) that are out of the scope of the article. Both section 4 and 5 could be shortened by avoiding the description of concepts that might fit for a book chapter, but not for a review article.
Response 1: Thanks for your careful review and constructive comments. According to your advice, we have revised the paragraphs in sections 4 and 5 to mitigate excessive elaboration. Please see the attached manuscript for the revised paragraphs.
Point 2: In fact, the crucial information of this article is reported in the tables, where DDSs are presented and summarized, making the paragraphs look dispersive and section 4 almost useless. Moreover, studies reported in the tables are not commented on which limitations authors found or problems linked to the use of those systems, which is a pivotal element when writing a review article. I think authors should replace most of the "basic" information in the paragraphs with a critical comment on the most important studies that they cite. I think authors should also consider changing perspective, and dividing DDSs not by application, but by type (NPs, hydrogels, implants ...) as they did in section 4. The article's title and focus are novel formulations for ocular delivery, so this should be the main theme of the whole work.
Response 2: Thank you for your valuable comments. To direct attention towards the notion of biodegradable ocular drug delivery systems and eliminate extraneous information, Sections 4 and 5 underwent revision wherein detailed fundamental content was omitted. In Section 4, concepts of ocular biodegradable formulations were introduced, with a focus on their applications for ocular delivery. Additionally, common obstacles encountered by these biodegradable formulations were also discussed. As per your recommendation, we have elaborated upon the implementation of various types of formulations by categorizing them according to the respective disease applications in Section 5. During the writing process, it became challenging to differentiate between various types of biodegradable formulations, as numerous studies integrate two different types of formulations. For instance, a hydrogel may serve as a vehicle that contains drug-loaded nanoparticles or use polymeric microneedles to deliver drug-loaded nanoparticles. Consequently, we aim to analyze distinct formulations and drug applications based on the original disease classification to align the primary focus of the article with the concept of “the formulation”. Besides, we also have highlighted the constraints and restrictions associated with the studies cited in the references. Please see the attached manuscript for the revised paragraphs.

Reviewer 2 Report
pharmaceutics-2222367
Updates on biodegradable formulations for ocular drug delivery in recent ten years
The manuscript by Tsung et al. summarized recent updates in biodegradable formulations for ocular drug delivery over the past decade. The authors also presented a comprehensive review of potential future trends in biodegradable ocular drug delivery systems and their potential for practical clinical applications. Below are some suggestions to improve the manuscript.
1. Similar reviews on nano formulations for ocular drug delivery are available. In the Introduction part, the authors should emphasize the novelty and contribution of this review in comparison with previous ones.
2. Section 4.1: the authors did not mention and discuss solid lipid nanoparticles and nanostructured-lipid carriers, which have been used in ocular delivery. These nanosystems are mentioned in some studies in tables 1 – 4.
3. How about the clinical status of nanosystems? Are there any nanosystems in clinical practice or clinical trials? The number might be small. If so, can the authors identify some challenges and suggest future directions for these nanosystems in ocular delivery? Please include them in section 4.
Author Response
Response to Reviewer 2 Comments
Point 1: Similar reviews on nano formulations for ocular drug delivery are available. In the Introduction part, the authors should emphasize the novelty and contribution of this review in comparison with previous ones.
Response 1: Thanks for your comments. In order to emphasize the novelty and contribution of this review, we revised the introduction section as follows:
……To the best of our knowledge, the majority of the extant literature on biodegradable DDSs examines a single system exclusively. In this vein, the present work stands out as the first comprehensive review to integrate the latest biodegradable ocular DDSs, whether currently on the market or in research. Each system is analyzed with regard to its design, potential for treating different eye diseases, potential challenges, and prospects for future development.
Point 2: Section 4.1: the authors did not mention and discuss solid lipid nanoparticles and nanostructured-lipid carriers, which have been used in ocular delivery. These nanosystems are mentioned in some studies in tables 1 – 4.
Response 2: Thanks for your valuable comments and suggestion. To supplement the discussion of solid lipid nanoparticles and nanostructured-lipid carriers, which were not explicitly referenced previously, lipid nanoparticles are addressed and discussed in Section 4.1.1. The revised section was as follows:
4.1.1. Nanoparticles
Nanoparticles (NPs) are colloidal particles with a spherical shape that are typically characterized by a size range spanning 1 to 1000 nm [43]. NPs are classified into two groups: polymeric NPs and lipid NPs. Polymeric NPs are solid colloidal particles made of non-biodegradable synthetic polymers or biodegradable macromolecular materials derived from synthetic, semisynthetic, or natural resources. Examples of natural polymers that can be used to make nanoparticles include chitosan, alginate, albumin, gelatin, and dextran. Biodegradable synthetic polymers include polylactic acid (PLA), polyglycolide (PGA), poly lactic-co-glycolic acid (PLGA), poly(L-lysine), polyaspartic acid, poly-alkyl cyanoacrylate, and polyethyleneimine (PEI) [36, 43]. Furthermore, mucoadhesive nanosystems that bind to mucins through hydrogen bonding and/or electrostatic interactions, thereby enhancing the residence time of NPs in ocular mucosal and anterior corneal tissues, are prepared using hydrophilic polymers such as polyethylene glycol (PEG) and polyvinylpyrrolidone (PVP) [44].
Polymeric NPs are hampered by cytotoxicity and a lack of suitable large-scale production techniques. In contrast, lipid NPs carry a much lower toxicological risk relying on non-toxic and biodegradable lipid components. Solid lipid nanoparticles (SLNs) and nanostructured lipid carriers (NLCs) are composed of a solid matrix that allows for controlled drug release, while also being more stable and cost-effective. SLNs utilize physiological lipids, eschew organic solvents, and enable large-scale production, thereby improving drug bioavailability, protecting sensitive drugs from harsh environments, and controlling drug release. NLCs are produced by controlling the mixing of solid lipids with liquid oil, which leads to special nanostructures in the matrix. The potential drawbacks of SLNs, such as limited drug-loading capacity and drug expulsion during storage, can be circumvented by the newer NLCs [45].
Drug-loaded NPs can be in the form of nanospheres, where a drug is evenly distributed over the polymer matrix, or nanocapsules, where a drug is enclosed within the polymer shell. Nanospheres are more stable and have higher drug loading capacity, while nanocapsules have better-targeted drug delivery and have the ability to protect the drug from degradation. According to current studies [46-50], drug-loaded NPs have the advantages of high drug retention, low dosing frequency, and low toxicity for the treatment of ocular surface diseases, and have the potential to replace conventional formulations as the main choice of anterior ocular disease treatment in the near future. However, although polymeric and lipid NPs have a strong carrying capacity for fat-soluble substances, their carrying capacity for water-soluble substances is insufficient, which is a direction that needs to be studied and improved.
Point 3: How about the clinical status of nanosystems? Are there any nanosystems in clinical practice or clinical trials? The number might be small. If so, can the authors identify some challenges and suggest future directions for these nanosystems in ocular delivery? Please include them in section 4.
Response 3: Thank you for your constructive comments. There are several nanosystems in ocular delivery that are currently in clinical practice or clinical trials. For example, commercial products such as Dexycu® and Durysta® are based on biodegradable polymers for ocular delivery. Additionally, there are several nanosystems in clinical trials, including Adeno-Associated Virus (AAV) vectors for gene therapy, liposome-based formulations, and nanoparticle-based formulations. Some of the key challenges in developing these nanosystems for ocular delivery include achieving sustained release, optimizing the pharmacokinetic and pharmacodynamic profiles, and minimizing toxicity and immunogenicity. Despite these challenges, the development of nanosystems for ocular delivery has shown great promise in improving drug efficacy, reducing the dosing frequency, and enhancing patient compliance, and further research in this area is warranted to address the unmet clinical needs in the field of ocular drug delivery. Following your advice, we included these challenges and discussed future directions for nanosystems in section 4 from lines 533-563, as follows:
The novel ONSs discussed above have been demonstrated to possess superior pharmacokinetic and pharmacodynamic properties compared to conventional therapies, as evidenced by their controlled release of therapeutic drugs, targeted delivery to specific tissues, extended contact time, prolonged circulation, enhanced penetration through biological barriers, and decreased elimination rates [72, 77, 78]. One of the major challenges is the optimization of drug loading and release, which could be achieved through the incorporation of various types of functional materials or the modification of the nanoparticle surface. Another important challenge is the improvement of the stability and biocompatibility of nanosystems, especially for long-term use. Additionally, the manufacturing process and scale-up production of nanosystems remain a challenge due to the complex synthesis procedures and the lack of cost-effective production techniques. Despite their potential benefits for ocular drug delivery, several challenges must be addressed to improve their efficacy and clinical feasibility. One of the major challenges is achieving adequate drug loading and release, which may be hindered by drug stability and solubility, as well as the properties of the nanosystem itself. Another obstacle is optimizing the size and surface properties of the nanosystem to ensure effective penetration and targeting in the eye while minimizing toxicity and immune reactions. Furthermore, regulatory and manufacturing challenges must be overcome to enable large-scale production and clinical translation of these complex drug delivery systems. Future research efforts in the field of ONSs could focus on several aspects to further enhance their efficacy and address current limitations. One such aspect is the development of more efficient and eco-friendly synthesis methods, which can improve the reproducibility and scalability of nanosystems. Advanced techniques such as 3D printing and microfluidics can also be explored to achieve this objective. Additionally, the design of nanocarriers, including size, shape, and surface properties, can be optimized to enhance their penetration across ocular barriers and improve their drug delivery efficiency. Co-delivery of multiple drugs or therapeutic agents using nanosystems can also be explored, as it has the potential to enhance therapeutic efficacy and reduce the frequency of administration. These research areas could lead to the development of more effective and targeted ocular drug delivery systems with reduced side effects and improved patient compliance.

Round 2
Reviewer 1 Report
I am glad that authors used my comments to reorganize their work and I see that the manuscript is now more centered on the actual focus of the review. I recommend it for publication.
Reviewer 2 Report
The manuscript was appropriately revised and can be accepted as is.